# Skin-lightening products and Jordanian women: Beliefs and practice. A cross-sectional study

**Manal Ayyash**[1]*, **Kamel Jaber**[2], **Razan I. Nassar**[3], **Leen Fino**[3], **Lana Mango**[2], **Alaa Abuodeh**[4]

1 Department of Pharmaceutics and Pharmaceutical Science, Faculty of Pharmacy, Applied Science Private University, Shafa badran, Amman, Jordan, 2 School of Medicine, The University of Jordan, Shafa badran, Amman, Jordan, 3 Department of Clinical Pharmacy and Therapeutics, Faculty of Pharmacy, Applied Science Private University, Shafa badran, Amman, Jordan, 4 Department of Pharmaceutical Chemistry and Pharmacognosy, Faculty of Pharmacy, Applied Science Private University, Shafa badran, Amman, Jordan

* m_ayyash@asu.edu.jo

## Abstract

**Data Availability Statement:** All relevant data are within the paper and Supporting Information files.

### Background

The use of skin-lightening products (SLPs) among Jordanian women has immensely increased and healthcare professionals have a vital role in raising public awareness of SLPs. The aim of this study is to identify SLPs practices among Jordanian women and their basic knowledge of the agents and the side effects associated with using these products.

### Methods

A cross-sectional study conducted during October to December of 2022. Jordanian women above 18 years of age were invited to participate via a survey link. Descriptive statistics were used, and logistic regression was applied to screen for variables affecting the knowledge score of the participants.

### Results

The mean age of the study participants (n = 384) was 32.04 (SD = 12.678). Results demonstrated that more than half of the participants (n = 193) reported current or past use of SLPs. Additionally, less than one-fifth (18.2%) of the participants (n = 70) reported previously experiencing some side-effects after using SLPs. About 90% of participants thought that these side-effects were caused by the active ingredients in SLPs. Most of the participants were able to identify some of the active ingredients used in SLPs such as Vitamin C (87.8%) and Hydroquinone (62.0%). It was also found that young participants, and those employed, or university students had higher knowledge scores of SLPs' active ingredients, and of their side-effects.

**Funding:** The authors received no specific funding for this work.

**Competing interests:** The authors have declared that no competing interests exist.

## Conclusion

This study demonstrated that Jordanian women are adequately informed about skin-lightening products. Moreover, the practices revealed an educated pattern of action when obtaining information regarding SLPs. Fundamentally, healthcare providers should be influential in educating consumers on the proper use. Strict guidelines and policies should target the practices concerned with these products.

## Introduction

The use of skin-lightening products (SLPs) has become a prominent practice amongst the general public around the world [1, 2]. Although both males and females partake in this practice; women have shown higher rates of involvement in all sorts of skin-whitening practices [3]. This might be a consequence of the social pressures applied to them [2]. As attaining a lighter skin complexion is sociologically and culturally believed to be associated with wealth and power in various parts of the world [2–4]. Moreover, pigmentary disorders such as melasma, periorbital melanosis, freckles, and post-inflammatory hyperpigmentation (PIH) can negatively impact women's quality of life [5–7]. Such factors have resulted in the availability of numerous therapeutic interventions such as skin whitening agents, chemical peels, and lasers [4]. Thus, there is a variation in the prevalence of the use of these skin-whitening approaches among women in different countries around the world [8–10]. Skin-lightening products (SLPs) are extravagantly available in the Jordanian market, e.g., products containing Hydroquinone, Arbutin, Kojic acid, and Glutathione. Laser skin treatment, and skin–peeling with its different mechanisms are also widely accessible in the Jordanian market [9].

Using SLPs without the appropriate directions and knowledge proved to be maleficent, causing patients different preventable side effects. For example, irritation, redness, and ochronosis (i.e. loss of elasticity of the skin and impaired wound healing) may be caused by the inappropriate use of Hydroquinone-containing preparations [11]. Thus, healthcare providers have the opportunity to raise public awareness and knowledge in the realm of skin bleaching [1, 2, 12].

The domain of skin-lightening practice among Jordanian women remains relatively unexplored. Moreover, products used to lighten skin can have very detrimental effects on the users' skin and overall health if used incorrectly. Therefore, this study aimed to assess the practices, knowledge, awareness, and perception among Jordanian women regarding the SLPs focusing on approaches utilized by the participants in their attempt to modify their skin tone. By addressing such aspects of knowledge and use, health of patients and users can be protected and improved by guiding proper usage.

## Methodology

### Study design, settings, and participants

The current descriptive cross-sectional study was conducted in the period between the 20[th] of October 2022 till the 28[th] of December 2022. An appropriate sample size of women was invited to participate in the study survey by sending the survey link through social media platforms (e.g., Facebook and WhatsApp). The inclusion criteria were women above 18 years old who reside in Jordan. Participants were informed about the aim of the study and that completing the survey would take approximately 7 minutes. Participants were informed that the participation is optional, and data gathered will be confidentially dealt with. Thereafter, participants

who wished to continue were asked to provide their electronic informed consent by clicking on the "agree" button, and for those who refused to participate, the "disagree" button.

## Survey instrument development and validation

A validated survey that assessed Jordanian women's knowledge, perception, uses, and beliefs on SLPs was developed based on a literature review that required validation Pharmacists and cosmeceutical experts assessed the survey's face and content validity. Experts' comments were collected and revised, and the survey was modified accordingly. The survey was piloted with a group of 20 individuals to improve clarity, readability of the survey's items, as well as to ensure its applicability to the targeted individuals. Cronbach's alpha coefficient was used to assess the internal consistency reliability. Following the pilot phase, irrelevant items were omitted, and similar items were combined. The survey consisted of four sections assessing different topics of interest. The survey was comprised of close-ended, multiple-choice questions, and 5-point Likert scale. The first section consisted of questions that collect participants' sociodemographic characteristics. The second section assessed women's perceptions toward SLPs using a 5-point Likert scale ranging from strongly agree to strongly disagree, while the following section explored the participants' practices and experiences with SLPs. The last section aimed to evaluate the participants' knowledge about SLPs, active ingredients, and the possible side effects.

## Ethical consideration

Ethical approval for the current study was obtained from the Ethics Committee at Applied Science Private University (Approval number: 2022-PHA-27). The participants were informed their participation in this study is voluntary. Also, the participants were informed that no identifying personal information is required in the study and that their anonymity is maintained, and gathered data will be dealt with confidentially, not to be shared or used for purposes outside the scope of the study.

## Sample size

Using the Epi Info program, the minimum representative sample size was calculated with a 95% confidence level, 50% expected frequency, 5% acceptable margin of error, and a design effect of 1.0. Three hundred and eighty-four participants were the minimum required number of participants.

## Statistical analysis

Once the required sample size was reached, the obtained participants' responses were coded and inserted into a customized database using the Statistical Package for the Social Sciences (SPSS), Version 24.0 (IBM Corp., Armonk, New York, USA). Qualitative variables were presented as frequency and percentages while continuous variables were presented as mean (standard deviation).

Multiple logistic regression was used to screen for factors affecting the use of sunscreen among the study participants. Logistic regression tests a model in order to predict a categorical outcome such as using sunscreen which was answered by the participants by 'Yes' or 'No'; on the other hand, the independent variables can be either continuous such as age, or categorical such as living place, educational level, employment, monthly income, marital status, smoking status, and exposure to high sunlight due to daily life activities and work [13]. As a first step, simple logistic regression was performed; any independent variable with a p-value <0.25 was considered suitable to be included in the multiple logistic regression. As a second step

(multiple logistic regression), any independent variable with a p-value ≤0.05 was considered significant. All independent variables were chosen after checking their independence, therefore, tolerance value of <0.2 and a variance inflation factor (VIF) of <5 were checked to guarantee that multicollinearity was not presented. Linear regression was conducted to screen for the variables affecting the knowledge level of the active ingredients used in skin whitening products, as well as the knowledge level of the side effects associated with using these products. Participants were asked about five active ingredients used in skin whitening products, thus, a score out of 5 was calculated. Likewise, they were asked about eight side effects associated with using skin whitening products, therefore, a score out of 8 was calculated.

## Results

A total of 389 participants were invited to fill out the survey. Nevertheless, five participants clicked on the *'disagree'* button indicating declining to participate in the study, therefore, they were excluded from the analysis, consequently, a total of 384 participants were included in the current study analysis, giving the study a response rate of 98.71%.

The mean age of the study participants (n = 384) was 32.04 (SD = 12.678). The majority of the participants were living in the central regions (95.6%), three-quarters of the participants held a bachelor's degree (75.0%), and 35.2% of the participants (n = 135) were employed. More than one-third of the study participants (n = 146) had a monthly income equal to or less than 250 JD per month. More than half of the participants were single (53.9%), and about 83.0% were non-smokers. Table 1 shows the detailed sociodemographic characteristics of the study's participants (n = 384).

### Participants' perceptions toward skin whitening products

Table 2 shows participants' perceptions toward skin lightening products. The highest agreement score was observed in *"There is low awareness about skin- lightening products in Jordan"* (82.1%), followed by *"Women are more concerned with the brand name and the product price rather than the active ingredient of the product"* (73.4%), and *"The availability and affordability of many skin- lightening products enhanced the purchase of these products"* (73.4%). Approximately, from 59% to 69% of the participants strongly agreed/ agreed on *"Women only use skin- lightening products for cosmetic purposes"* (68.8%), *"Women believe that using skin- lightening products is necessary"* (64.3%), *"Women believe that using skin- lightening products is safer than laser therapy"* (64.1%), and *"Women use skin- lightening products because they believe it would increase their marriage chances and get employed"* (58.9%). About half of the study participants strongly agreed/agreed *on "The use of skin- lightening products enhances women`s confidence and societal status"* (50.8%). Less than half of the study participants strongly agreed/agreed on *"The use of skin-lightening products should be restricted to medical purposes"* (44.8%), and *"Skin pigmentations cannot be completely removed"* (47.7%). *"All active ingredients used in skin-lightening products have the same efficacy"* had the lowest agreement score (14.0%).

### General lifestyle and practices among study participants

Fig 1 shows the natural skin tone (colour) among the study participants (n = 384). Most of the participants had a fair skin tone (46.1%, n = 177), followed by a light brown tone (38.0%, n = 146).

Table 3 shows the lifestyle and practices among study participants. Most of the participants (41.9%) were exposed moderately to high sunlight due to their life activities or work. Only about one-quarter of the study participants use sunscreen or sunblock products on regular

**Table 1. Sociodemographic characteristics of the study participants (n = 384).**

| Parameters | Mean (SD) | n (%) |
|---|---|---|
| **Age (years)** | **32.04 (12.678)** | |
| **Living place** | | |
| • North region (Irbid, Ajloun, Jerash, Mafraq) | | 5 (1.3) |
| • Central region (Amman, Zarqa, Balqa, Madaba) | | 367 (95.6) |
| • South region (Karak, Tafilah, Ma'an, Aqaba) | | 12 (3.1) |
| **Educational level** | | |
| • Secondary education or less | | 21 (5.5) |
| • Diploma | | 29 (7.6) |
| • Bachelor's degree | | 288 (75.0) |
| • Master's degree | | 33 (8.6) |
| • PhD | | 13 (3.4) |
| **Employment** | | |
| • Employed | | 135 (35.2) |
| • Un-employed | | 114 (29.7) |
| • Student | | 135 (35.2) |
| **Monthly income*** | | |
| • ≤ 250 JD/month | | 146 (38.0) |
| • 251–500 JD/month | | 91 (23.7) |
| • 501–750 JD/month | | 56 (14.6) |
| • 751–1000 JD | | 36 (9.4) |
| • > 1000 JD | | 55 (14.3) |
| **Marital status** | | |
| • Single | | 207 (53.9) |
| • Married | | 159 (41.4) |
| • Others (divorced or widowed) | | 18 (4.7) |
| **Smoking status** | | |
| • Smoker | | 60 (15.6) |
| • Ex-smoker | | 6 (1.6) |
| • Non-smoker | | 318 (82.8) |

*JD: 1 United States Dollar = 0.71 Jordanian Dinar (JD)

basis (n = 290). With regards to skin discolorations, freckles, and sunburns, 78.7% of the study participants reported previously experiencing them; largely, moderately, or partially (7.3%, 28.4%, and 43.0%, respectively). Approximately 28.0% of the participants (n = 107) reported allocating part of their income in order to purchase skin whitening products. Less than one-fifth of the study participants (18.0%, n = 69) reported currently using skin whitening products, while 32.3% had used them in the past, and 49.7% had never done so. Moreover, 31.8% of the study participants reported using homemade preparations (herbs, lemon) to whiten or lighten their skin (n = 122).

Regarding the factors that influence participants' choice of skin lightening products, three-quarters of the study participants (75.0%, n = 288) reported that the product cost influences their choice, while 25.0% reported that the cost has no influence. Moreover, 84.6% reported that the brand name influences their choice, on the other hand, 15.4% reported that the brand name had no influence. The same percentage was found with regard to the active ingredient (s), as 84.6% reported that the active ingredients influence their choice.

**Table 2. Participants (n = 384) perception toward skin lightening products.**

| Statement | Strongly agree n (%) | Agree n (%) | Neutral n (%) | Disagree n (%) | Strongly disagree n (%) |
|---|---|---|---|---|---|
| Women only use skin-lightening products for cosmetic purposes | 92 (24.0) | 172 (44.8) | 62 (16.1) | 51 (13.3) | 7 (1.8) |
| The use of skin-lightening products should be restricted to medical purposes | 73 (19.0) | 99 (25.8) | 109 (28.4) | 94 (24.5) | 9 (2.3) |
| The use of skin- lightening products enhances women's confidence and societal status | 82 (21.4) | 113 (29.4) | 102 (26.6) | 67 (17.4) | 20 (5.2) |
| All active ingredients used in skin-lightening products have the same efficacy | 12 (3.1) | 42 (10.9) | 105 (27.3) | 179 (46.6) | 46 (12.0) |
| Skin pigmentations cannot be completely removed | 44 (11.5) | 139 (36.2) | 122 (31.8) | 67 (17.4) | 12 (3.1) |
| Women are more concerned with the brand name and the product price rather than the active ingredient of the product | 115 (29.9) | 167 (43.5) | 81 (21.1) | 14 (3.6) | 7 (1.8) |
| Women believe that using skin-lightening products is necessary | 71 (18.5) | 176 (45.8) | 93 (24.2) | 40 (10.4) | 4 (1.0) |
| Women believe that using skin-lightening products is safer than laser therapy | 69 (18.0) | 177 (46.1) | 108 (28.1) | 27 (7.0) | 3 (0.8) |
| The availability and affordability of many skin-lightening products enhanced the purchase of these products | 81 (21.1) | 201 (52.3) | 83 (21.6) | 13 (3.4) | 6 (1.6) |
| There is low awareness about skin-lightening products in Jordan | 155 (40.4) | 160 (41.7) | 55 (14.3) | 12 (3.1) | 2 (0.5) |
| Women use skin-lightening products because they believe it would increase their marriage chances and get employed | 102 (26.6) | 124 (32.3) | 82 (21.4) | 54 (14.1) | 22 (5.7) |

## Knowledge about SLPs among study participants

Regarding sources of information used by the study participants to obtain information regarding skin whitening products (Fig 2), dermatologists had the highest percentage (79.4%), followed by community pharmacists (76.0%). On the other hand, television had the lowest score (9.4%).

Fig 3 displays the study participants' knowledge regarding active ingredients used in SLPs, as well as their sources of information regarding the active ingredients. When participants were asked which of the following ingredients are used in SLPs, 87.8% answered "*yes*" for vitamin c, 62.0% for Hydroquinone, 50.0% for Kojic acid, 37.0% for Glutathione, and 34.6% for Arbutin. Furthermore, the product's leaflet was the most reported source that the study participants used to know about the active ingredients of the product (76.3%), followed by social media (52.6%).

The majority of the participants believed that SLPs have side effects (88.3%, n = 339). Additionally, 18.2% of the participants (n = 70) previously experienced some side-effects after using the SLPs.

When participants were asked which of the following side effects may arise as a result of using SLPs (Fig 4); skin irritation (93.0%), followed by skin redness (90.9%), and acne (71.6%) were the most reported side effects.

Participants were surveyed regarding their reaction towards SLPs subsequent occurrence of side-effects 242 participants (63.0%) responded that they would stop using the product immediately and consult a dermatologist/pharmacist/beauty expert, 91 participants (23.7%) would stop using the product immediately and never use it again, 50 participants (13.0%) would stop using the product immediately and purchase a different product, and only one participant (0.3%) would continue using the product.

With regards to the reasons behind the side-effects of the SLPs, 90.1% (n = 346) thought that the active ingredient(s) would be the reason, 74.2% (n = 285) answered "*Yes*" for the

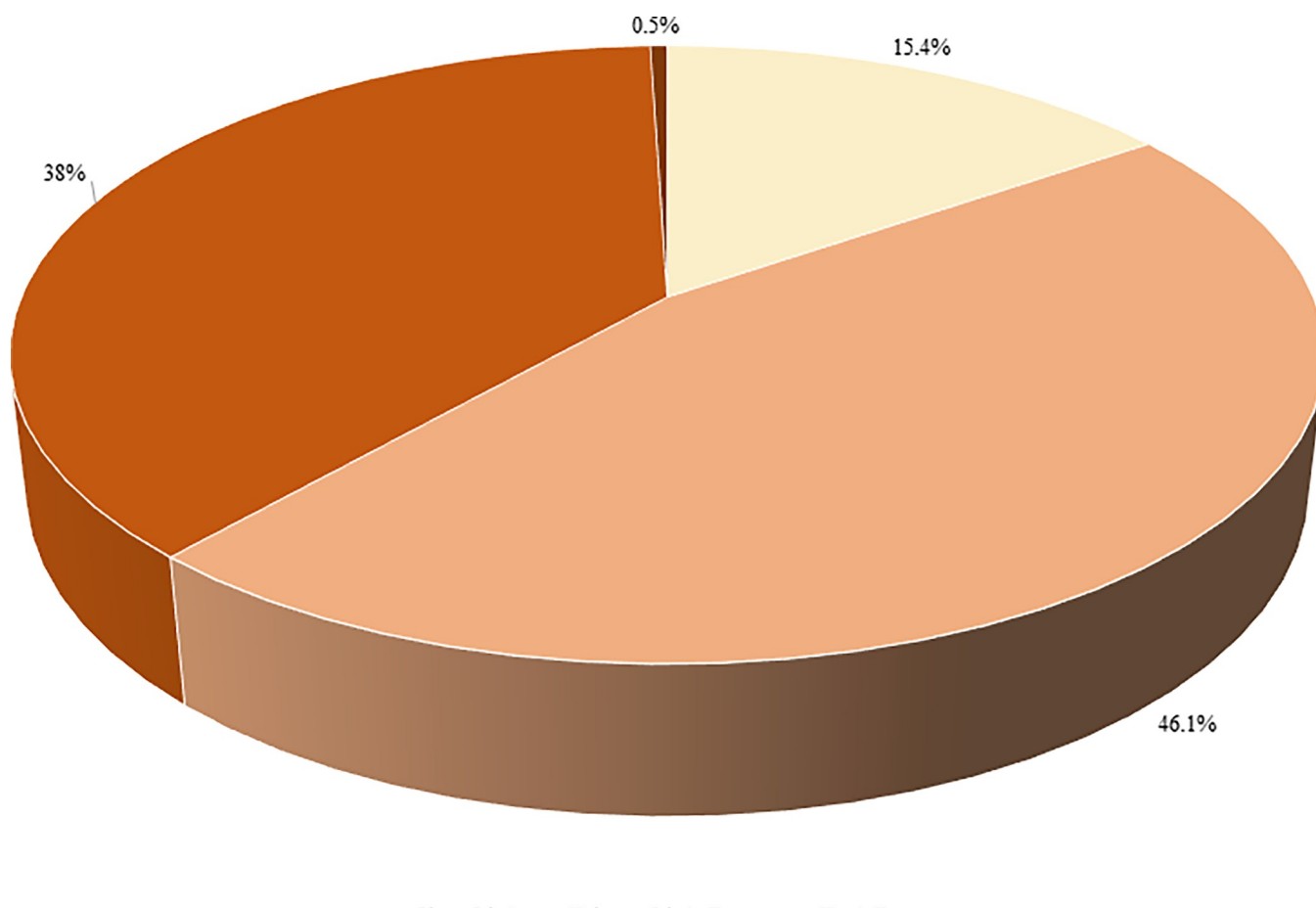

**Fig 1. Natural skin tone (colour) among the study participants (n = 384).**

improper time of use of the product, 85.4% (n = 328) for the excessive use of the product. 89.8% (n = 345) for the lack of information on how to use the product, and 88.5% (n = 340) for using more than one product at the same time.

The most preferred skin-lightening method reported by the study participants was peeling (37.5%, n = 144), followed by SLPs (37.0%, n = 142), and laser (14.6%, n = 56). About 11.0% of the participants (n = 42) preferred other methods.

More than half of the participants (54.2%, n = 208) said that they would use the SLPs if their dermatologist recommended them. Approximately 88.0% (n = 337) of the study participants reported that they have used or would use the SLPs in order to remove any skin pigments (e.g., melasma, dark sport, and freckles), on the other hand, 12.2% (n = 47) reported the opposite. Additionally, about half of the participants (49.5%, n = 190) reported that they would use the products or have used them to lighten their skin for aesthetic purposes only, and 194 partici-pants (50.5%) reported the opposite. More than 70% of the participants (n = 281) reported that these products can be applied on the face and other parts of the body, however, 26.8% (n = 103) reported that they can only be applied on the face.

Creams, and serum were the two forms that the study participants preferred the most. On the other hand, the injection was the least preferred form (Fig 5).

**Table 3. General lifestyle and practices among study participants (n = 384).**

| Questions | n (%) |
|---|---|
| Does your daily life activities/work predispose you to high sunlight exposure? | |
| • Yes, largely | 43 (11.2) |
| • Yes, moderately | 161 (41.9) |
| • Yes, partially | 129 (33.6) |
| • No | 51 (13.3) |
| Do you regularly use sunscreen or sunblock products? | |
| • Yes | 290 (24.5) |
| • No | 94 (75.5) |
| Have you ever had skin discolorations such as melasma, freckles or sunburn? | |
| • Yes, largely | 28 (7.3) |
| • Yes, moderately | 109 (28.4) |
| • Yes, partially | 165 (43.0) |
| • No | 82 (21.4) |
| Do you allocate part of your income for purchasing skin-lightening products? | |
| • Yes | 107 (27.9) |
| • No | 277 (72.1) |
| Do you use skin- lightening products? | |
| • Yes, I use it currently | 69 (18.0) |
| • Never used it | 191 (49.7) |
| • I used it before | 124 (32.3) |
| Have you ever tried using home-made preparations (herbs, lemon) to whiten/lighten your skin? | |
| • Yes | 122 (31.8) |
| • No | 262 (68.2) |

## Assessment of factors affecting the use of sunscreens regularly among study participants

Multiple logistic regression analysis of factors affecting the use of sunscreens regularly revealed that participants' age, education level, employment, and exposure to high sunlight due to daily life activities or work significantly affected the use of sunscreens regularly (Table 4). Thus, being younger (OR = 0.961, P-value = 0.002), as well as holding less than a bachelor's degree or higher (OR = 0.273, P-value = <0.001), are significantly associated with using sunscreens regularly. Furthermore, working or being a university student (OR = 2.068, P-value = 0.042), or being exposed to high sunlight due to daily life activities or work (OR = 1.316, P-value = 0.012) is significantly associated with using sunscreens regularly.

## Assessment of factors affecting the knowledge levels of active ingredients and the side effects associated with using SLPs among study participants

According to the linear regression analysis, participants' ages significantly influenced their knowledge level of the active ingredients used in SLPs (Beta = -0.012, P-value = 0.003) and the side effects associated with using these products (Beta = -2.484, P-value = 0.013), with younger participants having higher knowledge scores. In addition, being employed or a university student, thus leaving home, was significantly associated with a higher active ingredients' knowledge score (Beta = 0.325, P-value = 0.005), as well as a higher side effects score (Beta = 3.170, P-value = 0.005).

**Fig 2. Sources of information used by the study participants to obtain information regarding SLPs.**

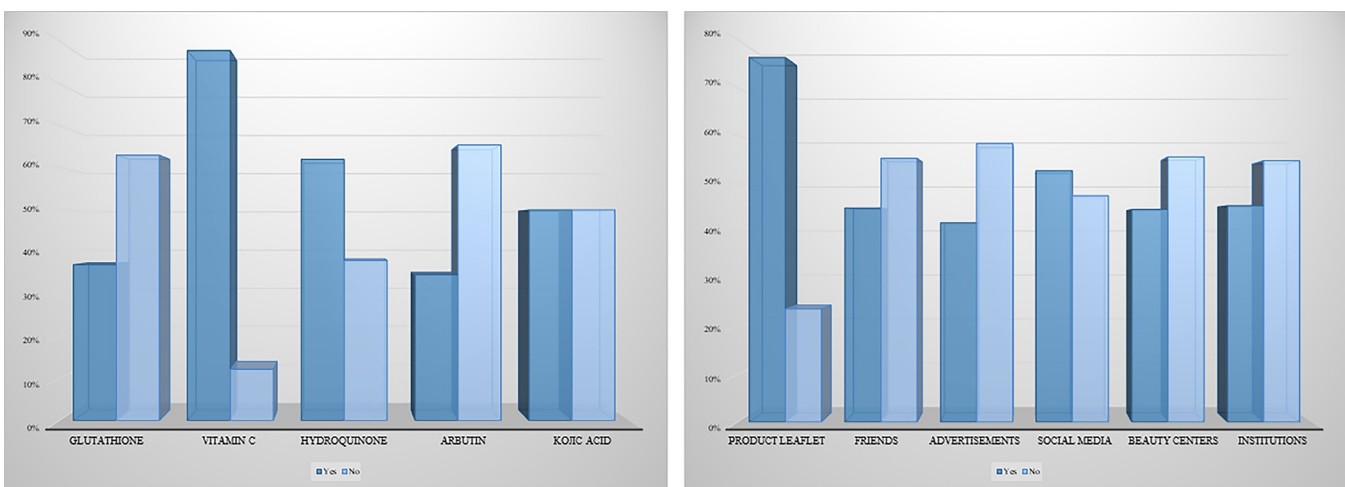

**Fig 3. Participants' knowledge regarding active ingredients used in SLPs, as well as their sources of information regarding the active ingredients.**

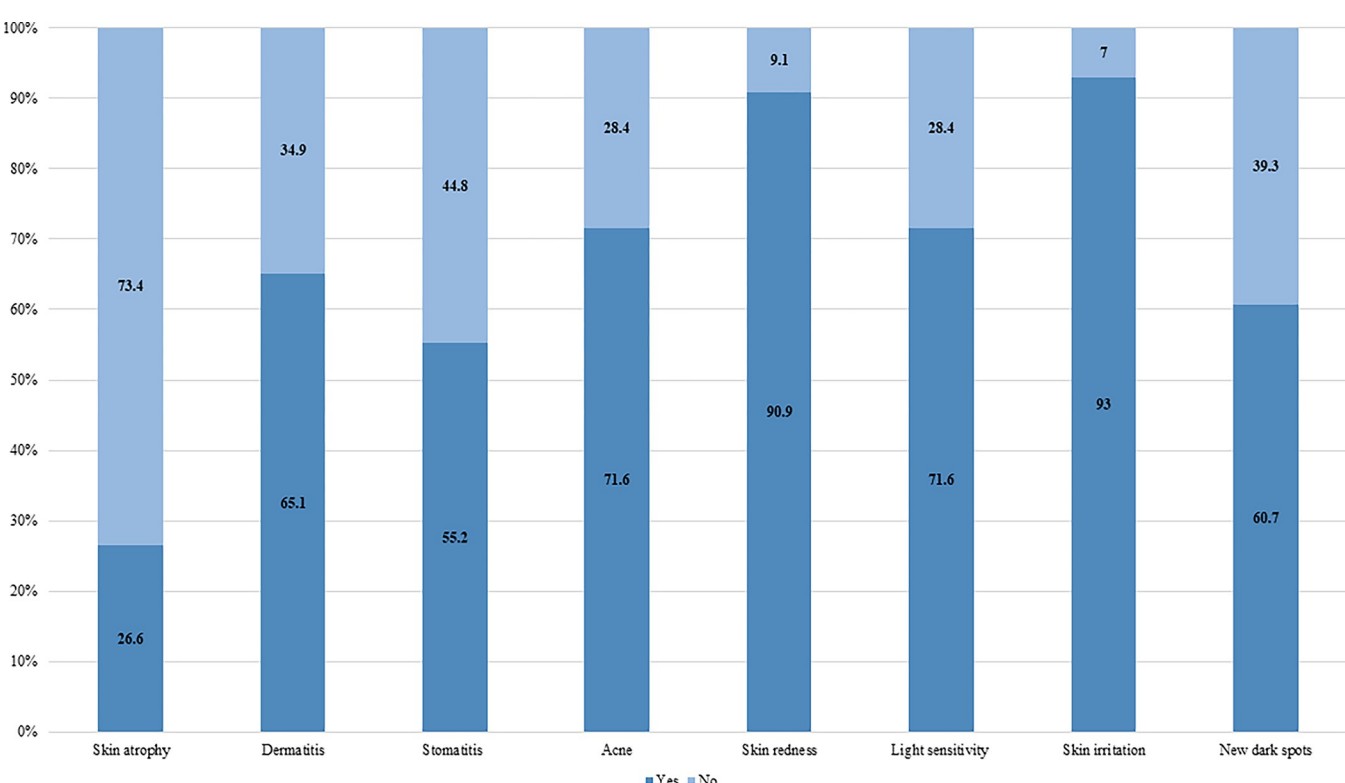

**Fig 4. Study participants' knowledge regarding SLPs side effects.**

## Discussion

The current study was conducted to assess skin-lightening products practices among Jordanian women. More than half of the study participants reported current or past use of SLPs. About 18% of the participants experienced previous side effects after using the SLPs, and around 90% of participants thought that these side effects were caused by the active ingredients in the SLPs. Most of the participants were able to identify some of the active ingredients used in SLPs such as vitamin c, Hydroquinone, and Kojic acid. More than 80% of the participants believed that there is a low awareness level about skin-lightening products among Jordanian women in general. Moreover, it was found that young participants, and those employed, or university students had higher knowledge scores of the active ingredients used in SLPs, as well as the side effects associated with using these products.

### Perceptions of skin-lightening products (SLPs)

Almost 70% of the participants believed that women only use SLPs for cosmetic purposes. This is consistent with the results of a cross-sectional study conducted in Saudi Arabia–Jordan's neighbouring country–where it was found that the main reason for using SLPs was cosmetic purposes (66.0%). According to the study conducted in Saudi Arabia, 66.6% of women perceived lighter skin tones as more beautiful. This demonstrates a possible regional pattern in the mentality surrounding SLPs. On the other hand, a similar study conducted in Somalia revealed that 12.6% agreed that lighter skin tones make women look younger, and only 14.2% of Somalis considered lighter skin tones as more youthful and beautiful [12].

In 2010, a study was conducted in Jordan to investigate the factors that influence the use of SLPs, as well as their awareness of these products. It was found that 60.7% of the women use

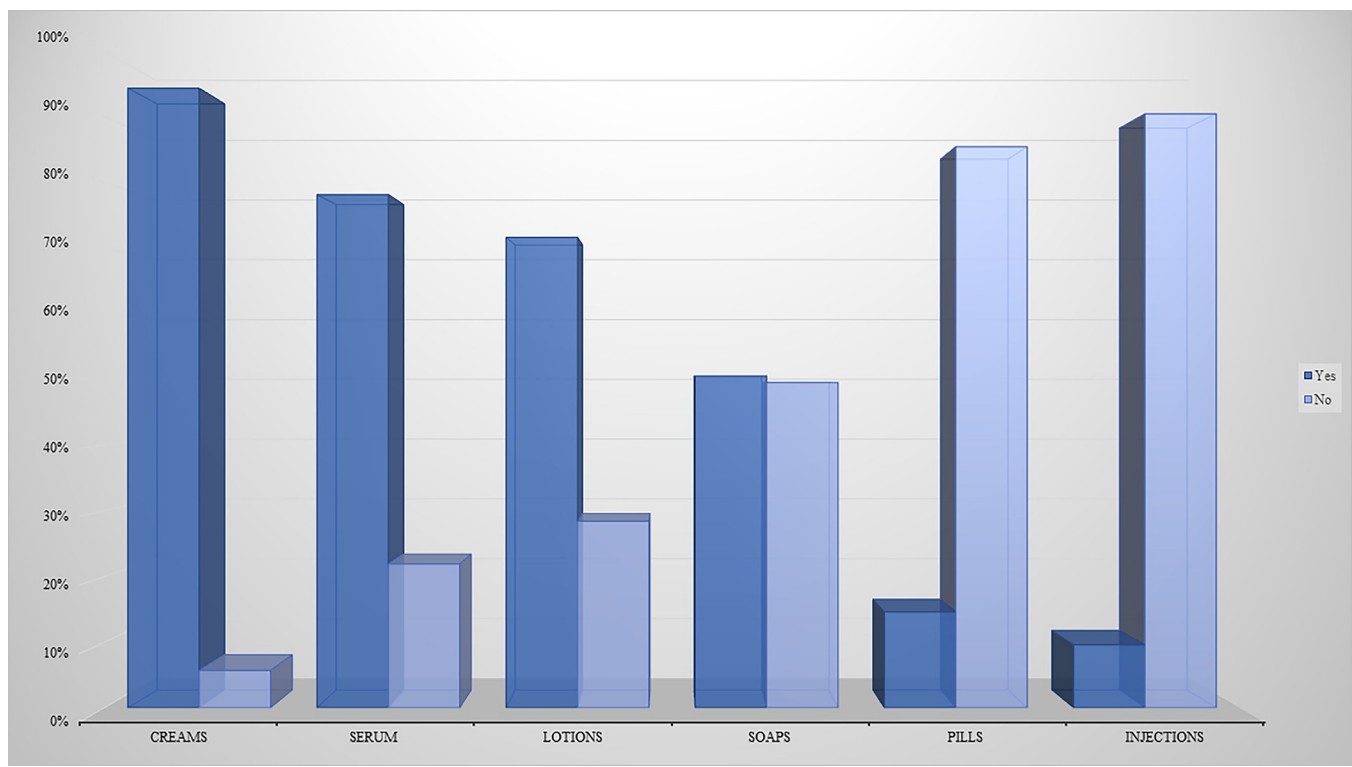

**Fig 5. Preferred SLPs' forms among study participants (n = 384).**

SLPs, furthermore, 62.3% of the women believed that a lighter tone is more beautiful. In the current study, a higher percentage (70%) supports the belief of this beauty-driven pattern of SLPs use [9].

The correlation between lighter skin tone and women's perception of beauty emphasizes the psychosocial factors that play a significant role in SLPs' use. The study results showed that around half of the women agreed that having a lighter skin tone gives women more confidence and social status. This finding is similar to the results of the study conducted in Saudi Arabia, as 69.9% of the women thought that a woman with lighter skin tone is assumed to be from a higher social class [10].

With regard to job opportunities, 58.9% of the women agreed that having a lighter skin tone improves a woman's chances of getting employed. This supports the strikingly similar findings from the Jordanian study that was conducted in 2010 [9]. Women in the current study believed that using SLPs also increases women's chances of getting married; so did the participants in the Saudi Arabia study [10]. Women may feel burdened by this because they would use SLPs to lighten their skin tone in an effort to conform to social standards of beauty in order to feel accepted by society.

## General lifestyle and practices among study participants

It is worth mentioning that 78.7% of the study participants reported previously experiencing skin discolorations, freckles, and sunburns. However, only about one-quarter of the participants used sunscreen or sunblock regularly, despite the moderate-to-high sun exposure that 41.9% of participants experience during the day. In contrast, 86% of Asian Indian women reported using sun protection measures such as sunscreen [14]. This wide gap in using

**Table 4. Assessment of factors affecting the use of sunscreens regularly among study participants (n = 384).**

| Parameter | Use of sunscreen regularly [0: No, 1: Yes] | | | |
|---|---|---|---|---|
| | OR | P-value[#] | OR | P-value[$] |
| **Age**** | **0.957** | **<0.001**[^] | **0.961** | **0.002*** |
| | | | Lower CI for OR = 0.937 Upper CI for OR = 0.985 | |
| Living place • North or South regions (n = 17, 4.4%) • Central regions (n = 367, 95.6%) | Reference 1.539 | 0.506 | —— | —— |
| Educational level • < Bachelor's degree (n = 50, 13.0%) • ≥ Bachelor's degree (n = 334, 87%) | Reference 0.236 | **<0.001**[^] | 0.273 | **<0.001**[^] |
| | | | Lower CI for OR = 1.774 Upper CI for OR = 7.580 | |
| Employment–leaving home • Un-employed (n = 114, 29.7%) • Employed and students (n = 270, 70.3%) | Reference 0.523 | **0.009**[^] | 2.068 | **0.042*** |
| | | | Lower CI for OR = 0.240 Upper CI for OR = 0.975 | |
| Monthly income • ≤ 500$/month (n = 237, 61.7%) • > 500$/month (n = 147, 38.3%) | Reference 0.943 | **0.810** | 0.658 | 0.140 |
| | | | Lower CI for OR = 0.871 Upper CI for OR = 2.650 | |
| Marital status • Single, divorced, or widowed (n = 225, 58.6%) • Married (n = 159, 41.4%) | Reference 2.676 | **<0.001**[^] | 1.664 | 0.113 |
| | | | Lower CI for OR = 0.320 Upper CI for OR = 1.127 | |
| Smoking status • Non-smoker and ex-smoker (n = 324, 84.4%) • Smoker (n = 60, 15.6%) | Reference 1.147 | 0.668 | —— | —— |
| High sunlight exposure due to daily life activities and work • No (n = 51, 13.3%) • Yes (n = 333, 86.7%) | Reference 1.301 | **0.004**[^] | 1.316 | **0.012*** |
| | | | Lower CI for OR = 1.208 Upper CI for OR = 4.781 | |

** Continuous independent variable

# Using simple logistic regression

$ Using multiple logistic regression

^ Eligible for entry in multiple logistic regression (significant at 0.25 significance level)

*Significant at 0.05 significance level

CI: Confidence interval

sunblock and sunscreen between women in Jordan and India could be due to the varying skin tones commonly found in Jordan as 46.1% of participants have a fair skin tone and 38% have a light brown skin tone Which might be an explanation of the underutilization of sun protection measures in Jordan.

On the other side of the globe, a study conducted in the USA showed that 36.6% of the participants were unaware that the sun was a possible trigger or aggravating factor for pigmentation [5]; this alarming lack of public awareness may also explain the lack of use of these products in Jordan.

With regards to SLPs' use, only one-fifth of the present study participants reported current use of SLPs, and 32% were using them in the past; however, a higher percentage (60.7%) was shown in the study that was conducted in Jordan in 2010 [9]. Different percentages were reported in other countries; for example, in Saudi Arabia, two-thirds of the women (63.1%) reported using SLPs [10]. In Africa—namely South Africa, Somalia, and Sudan—there was a noticeable variation in the use of SLPs ranging from 30% to 75% [12, 15, 16].

The study results revealed that 31.8% of participants resorted to homemade herbal preparations to whiten or lighten their skin tone. A similar method of choice has been seen for 48% of Sudanese women [16].

According to the study's findings, the product's brand name was found to be the most influential factor in regard to purchasing SLPs, followed by the SLP's active ingredients, and the product's cost. These findings demonstrate the significance of branding and marketing in influencing the purchasing decisions of the general public. It was observed that among women from South Africa and Malaysia, the brand name was the factor that affected purchases of skin lightening products the most, followed by the price and then active ingredients [3, 15].

## Knowledge about SLPs among study participants

With regards to the sources of information regarding SLPs, television was the least used source. This can be explained by the fact that many studies stated that the way SLPs are advertised on television sells and emphasizes the idea of the superior beauty of lighter skin tones, urging the audience to purchase those products [9, 10, 16, 17].

Word of mouth was reported to be another source of reliable information by around two-thirds of the current study participants. Other studies have examined the impact of recommendations received from friends and family members, which some consider to be a significant and reliable indicator of a product's efficacy [12, 16].

Most of the study participants reported using the product leaflet to obtain information on SLPs, this reflects the participant's understanding of the possible effects of these ingredients on their cosmetic appearance and overall health. However, 20% of Saudi women were willing to use ambiguous bleaching products, which is extremely concerning [1].

A great majority of the participants know that vitamin c and hydroquinone are used in SLPs, and only a minority were aware of the use of other active ingredients such as kojic acid, glutathione, and arbutin. This shows that despite the participants' ability to identify a few of the active ingredients found in SLPs, they are still unaware of others, equally common ingredients used in a wide variety of SLPs that are present in the market.

With the availability of such products in the market, it would be expected that the vast majority of the women knew that SLPs can have side effects (88.3%); an improvement from the 62.5% back in 2010 [9]. This might pave the way for enacting policies and health measures with regard to SLPs use, which in turn will improve the product experience and encourage a much safer practice. Only 18.2% of participants reported experiencing side effects as a result of using SLPs. In a study of Pakistani women, side effects from SLPs were reported by 60% of the participants [18]. Among Saudi women, an equally high percentage of participants (62.3%) experienced side effects [10]. Healthcare providers should warn their patients of the possible side effects of SLPs and inform them to seek help if these side effects arise. Moreover, practitioners in the healthcare as well as skincare industries should promote SLPs while also being transparent such as by outlining the possible side effects of these products.

Most participants believed that skin irritation, skin redness, and acne are the most likely side effects associated with using SLPs. With regards to this, other studies in certain African regions reported a recurrence of these side effects among the study participants; the most

common side effects were skin atrophy, acne vulgaris, and allergic contact dermatitis [12, 19]. Telangiectasia was most commonly reported by Middle Eastern women [10].

As for the site of application of SLPs, more than 70% of the participants reported that these products can be applied on the face and other parts of the body, however, 26.8% reported that the products can be applied on the face only. In the study which was conducted among Jordanian women, participants reported applying these products mostly on the face (81.9%) rather than other body parts, which was also consistent among Saudi and Somali women [10, 12].

According to the study results, SLP was the most preferred method to lighten the skin after peeling. Thus, skin-lightening via SLPs has become a common practice among Jordanian women, making it worthwhile to educate the public about the mechanism of action of the products, their proper use, and their possible side effects.

Among women in Saudi Arabia, SLPs in cream formulation was reported as the most common form used by participants (63%) [10]. Similar findings were revealed among the current study participants, as creams and serums were the two forms that were preferred the most as opposed to the injectable form. In contrast, the study in Sudan showed that only 30% of the study participants preferred topical cream formulation [16].With regards to other forms, such as soap, more than three-quarters of Sudanese women who use SLP preferred the soap form.A lower percentage was observed in the current study, as only half of the women preferred to use the soap form.

According to the regression analysis, younger participants showed a higher knowledge of SLPs active ingredients, as well as the side effects. Another study conducted in the Middle East showed that the youth nowadays have significant knowledge with regards to SLPs due to the limitless advertisement of these products on social media [20]. Young women seem eager to obtain the best product that suits their skin with least possible side effects. Hence, indulging in acquiring the details of the product's composition as well as, of course, its risky potential.

## Limitations

Despite efforts made to include participants from different Jordanian socioeconomic statuses and locations in this study. The limited accessibility might have rendered this study's conclusions generalizability, and they might not accurately reflect Jordanian women's practices regarding SLPs. Recall bias is also a possibility as self-reporting was cardinal methodology in this study.

## Conclusions

Skin-lightening practices are aspired by women of different communities and cultures, driven by a variety of reasons that may be cosmetic, social or psychological. This practice is one that should be regulated and monitored by authorizing forces and health institutions of concern in Jordan to prevent the hazards which could arise if these products are obtained or consumed casually. The role of specialists is crucial starting from dermatologists to whom women reach out if they suffered any skin pigmentary disorders, to the community pharmacists who are responsible for dispensing SLPs. Healthcare providers should be influential in educating the patients or consumers on the proper use, the hazards accompanying and how to respond to these health threats. This investigation can serve as a scaffold for instituting health policies and regulations that guide such use, and force strict guidelines on where and how these products can be obtained, moreover, these policies should target the current giant wave of social media since it has substantial potential to present these products without much control over it. Campaigns and education programs for women should be undertaken, university courses and

workshops can also serve as basis for knowledge, one that can be incorporated into the curricula of numerous specialties, or even as optional courses in which students may enrol.

## Supporting information

**S1 Fig.**
(ZIP)

**S1 File.**
(PDF)

**S2 File.**
(RAR)

**S3 File.**
(RAR)

## Acknowledgments

The authors have no acknowledgments to be mentioned.

## Author Contributions

**Conceptualization:** Manal Ayyash, Kamel Jaber, Leen Fino, Lana Mango, Alaa Abuodeh.

**Data curation:** Razan I. Nassar.

**Formal analysis:** Razan I. Nassar.

**Investigation:** Manal Ayyash, Kamel Jaber, Leen Fino, Lana Mango, Alaa Abuodeh.

**Methodology:** Manal Ayyash, Kamel Jaber, Razan I. Nassar, Leen Fino, Lana Mango, Alaa Abuodeh.

**Project administration:** Manal Ayyash, Kamel Jaber, Leen Fino, Lana Mango.

**Software:** Kamel Jaber.

**Supervision:** Manal Ayyash.

**Validation:** Manal Ayyash, Razan I. Nassar.

**Visualization:** Razan I. Nassar, Alaa Abuodeh.

**Writing – original draft:** Manal Ayyash, Kamel Jaber, Razan I. Nassar, Leen Fino, Lana Mango.

**Writing – review & editing:** Manal Ayyash, Kamel Jaber, Razan I. Nassar, Leen Fino, Lana Mango, Alaa Abuodeh.

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
