## [Decision Letter · Decision Letter 0]

14 Sep 2023

PONE-D-23-24948Skin-Lightening Products and Jordanian Women: Beliefs and Practice. A Cross-sectional Study PLOS ONE

Dear Dr. Ayyash,

Thank you for submitting your manuscript to PLOS ONE. After careful consideration, we feel that it has merit but does not fully meet PLOS ONE’s publication criteria as it currently stands. Therefore, we invite you to submit a revised version of the manuscript that addresses the points raised during the review process.

We look forward to receiving your revised manuscript.

Kind regards,

Nicholas Aderinto Oluwaseyi

Academic Editor

PLOS ONE

Journal Requirements:

Reviewers' comments:

Reviewer's Responses to Questions

**Comments to the Author**

1. Is the manuscript technically sound, and do the data support the conclusions?

Reviewer #1: Yes

Reviewer #2: Yes

2. Has the statistical analysis been performed appropriately and rigorously? 

Reviewer #1: Yes

Reviewer #2: Yes

3. Have the authors made all data underlying the findings in their manuscript fully available?

Reviewer #1: Yes

Reviewer #2: Yes

4. Is the manuscript presented in an intelligible fashion and written in standard English?

Reviewer #1: Yes

Reviewer #2: Yes

5. Review Comments to the Author

Reviewer #1: Did you do a pilot study to validate the questionnaire? If so, please include the study's findings and the adjustments made to the instrument.

Alos, please include the significance of your study clearly

Reviewer #2: The authors conducted a well-structured cross-sectional study to assess the perceptions, practices, and knowledge of Jordanian women on the use of skin-lightening products (SLP). After a thorough review of the manuscript, the following are comments/suggestions from my end.

1.The methods section of the abstract needs details on study duration, inclusion and exclusion criteria, and type of analysis performed (instead name of the software).

2.On what basis the participants were invited through social media platforms? I mean how did you choose a person to send the link (apart from the inclusion criteria). The survey link was not posted on social media for an open response?

3.Line No. 80. “Any independent variable with a p-value <0.25 was considered suitable to be included in the multiple logistic regression”. What is the reference for this cut-off value?

4.Lines No. 135 & 136 and 142 & 143: Both sentences have information on the “source of information” with different findings. Needs clarity on this.

5.Lines No. 175 to 178: The risk factors mentioned here are not appropriately interpreted as the ORs of all these associations are not on one side (either less than 1 or more than 1).

6.Table 4: What are the groups used for OR calculation for the age parameter (similar to other parameters given)?

7.Table 4: 95% Confidence Intervals required along with OR for better interpretation of results.

8.Table 4: Better to mention the number with percentage for each group.

6. PLOS authors have the option to publish the peer review history of their article (what does this mean?). If published, this will include your full peer review and any attached files.

Reviewer #1: **Yes: **Mohamed Rashrash

Reviewer #2: **Yes: **Krishna Undela

---

## [Author Response · Author response to Decision Letter 0]

6 Oct 2023

To Reviewers

Response to reviewer #1:

C1. Did you do a pilot study to validate the questionnaire? If so, please include the study's findings and the adjustments made to the instrument.

Also, please include the significance of your study clearly.

R1 A. We conducted a pilot study prior to survey distribution. We included it within the methodology section under the subheading ‘Survey instrument development and validation’. Lines 59-62.

“The survey was piloted with a group of 20 individuals to improve clarity and readability of the survey’s items, as well as to ensure its applicability to the targeted individuals. Cronbach's alpha coefficient was used to assess the internal consistency reliability. Following the pilot phase, irrelevant items were omitted, and similar items were combined.”

R1 B. The significance of the study was expressed in a more enunciated manner addressing the risks of such products and the importance of guiding proper use. Lines 39 and 40 & lines 43 and 44.

Response to reviewer #2:

C1. The methods section of the abstract needs details on study duration, inclusion and exclusion criteria, and type of analysis performed (instead name of the software).

R1. The methodology section of the abstract was modified including data on inclusion criteria and time settings of the study. Lines 6-8.

C2. On what basis the participants were invited through social media platforms? I mean how did you choose a person to send the link (apart from the inclusion criteria). The survey link was not posted on social media for an open response?

R2. The survey link was sent directly to participants who may fulfill the inclusion criteria (Jordanian women above 18) via WhatsApp which recruited majority of respondents. Thereafter, the link was sent via Facebook messages to further women. Of course, we made sure that whoever to fill the survey should be living in Jordan as the survey clearly states in its introduction. Also, the first section of the questionnaire includes a question stating the place of residence of the participants which made sure only those living in Jordan are among the sampled population. The link was not posted to groups as that was perceived to target a more random population.

C3. Line No. 80. “Any independent variable with a p-value <0.25 was considered suitable to be included in the multiple logistic regression”. What is the reference for this cut-off value?

R3. The reference to this cut-off value is based on ‘Tabachnick B, Fidell L. Multivariate regression. Using Multivariate Statistics (5th ed) Boston, MA: Pearson Education. 2007:117–59’.

C4. Lines No. 135 & 136 and 142 & 143: Both sentences have information on the “source of information” with different findings. Needs clarity on this.

R4. Adjustments were applied to the manuscript to resolve possible confusion. Sources mentioned in lines 135 and 136 are regarding sources of information on the skin lightening product as a whole, while lines 142 and 143 are meant with the knowledge of the active ingredients within the product which was properly stated in the corresponding locations.

C5. Lines No. 175 to 178: The risk factors mentioned here are not appropriately interpreted as the ORs of all these associations are not on one side (either less than 1 or more than 1).

R5. The sentence was edited as follows, making sure that all the ORs are on one side: “Thus, being younger (OR= 0.961, P-value= 0.002), as well as holding less than a bachelor’s degree or higher (OR= 0.273, P-value= <0.001), are significantly associated with using sunscreens regularly. Furthermore, working or being a university student (OR= 2.068, P-value= 0.042), or being exposed to high sunlight due to daily life activities or work (OR= 1.316, P-value= 0.012) is significantly associated with using sunscreens regularly”.

C6. Table 4: What are the groups used for OR calculation for the age parameter (similar to other parameters given)?

R6. - According to the definition of logistic regression found in the following book (Pallant, Julie. SPSS Survival Manual: a Step by Step Guide to Data Analysis Using SPSS.)/ Chapter 14

“Logistic regression allows you to test models to predict categorical outcomes with two or more categories. Your predictor (independent) variables can be either categorical or continuous, or a mix of both in the one model.” 

Therefore, there were no categories for age, as it was a continuous variable. 

- To clarify the previous point; the following sentence: “The logistic regression was used to screen for the independent variables (age, living place, educational level, employment, monthly income, marital status, smoking status, and exposure to high sunlight due to daily life activities and work) affecting the use of sunscreen among the participants” … was change to “Multiple logistic regression was used to screen for factors affecting the use of sunscreen among the study participants. Logistic regression tests a model in order to predict a categorical outcome such as using sunscreen which was answered by the participants by ‘Yes’ or ‘No’; on the other hand, the independent variables can be either continuous such as age, or categorical such as living place, educational level, employment, monthly income, marital status, smoking status, and exposure to high sunlight due to daily life activities and work”

- The book reference was added to the reference 

- Furthermore, below Table 4, it was clarified that age is a continuous variable. 

C7. Table 4: 95% Confidence Intervals required along with OR for better interpretation of results.

R7. Done. The confidence intervals were added to Table 4

C8. Table 4: Better to mention the number with percentage for each group.

R8. Done. The percentages were added to Table 4

---

## [Decision Letter · Decision Letter 1]

23 Oct 2023

Skin-Lightening Products and Jordanian Women: Beliefs and Practice. A Cross-sectional Study

PONE-D-23-24948R1

Dear Dr. Ayyash,

We’re pleased to inform you that your manuscript has been judged scientifically suitable for publication and will be formally accepted for publication once it meets all outstanding technical requirements.

Kind regards,

Nicholas Aderinto Oluwaseyi

Academic Editor

PLOS ONE

Additional Editor Comments (optional):

Reviewers' comments:

Reviewer's Responses to Questions

**Comments to the Author**

1. If the authors have adequately addressed your comments raised in a previous round of review and you feel that this manuscript is now acceptable for publication, you may indicate that here to bypass the “Comments to the Author” section, enter your conflict of interest statement in the “Confidential to Editor” section, and submit your "Accept" recommendation.

Reviewer #1: All comments have been addressed

Reviewer #2: All comments have been addressed

2. Is the manuscript technically sound, and do the data support the conclusions?

Reviewer #1: Yes

Reviewer #2: Yes

3. Has the statistical analysis been performed appropriately and rigorously? 

Reviewer #1: Yes

Reviewer #2: Yes

4. Have the authors made all data underlying the findings in their manuscript fully available?

Reviewer #1: Yes

Reviewer #2: Yes

5. Is the manuscript presented in an intelligible fashion and written in standard English?

Reviewer #1: Yes

Reviewer #2: Yes

6. Review Comments to the Author

Reviewer #1: (No Response)

Reviewer #2: (No Response)

7. PLOS authors have the option to publish the peer review history of their article (what does this mean?). If published, this will include your full peer review and any attached files.

Reviewer #1: No

Reviewer #2: **Yes: **Krishna Undela

---

## [Editor Report · Acceptance letter]

10 Nov 2023

PONE-D-23-24948R1 

Skin-Lightening Products and Jordanian Women: Beliefs and Practice. A Cross-sectional Study 

Dear Dr. Ayyash:

I'm pleased to inform you that your manuscript has been deemed suitable for publication in PLOS ONE. Congratulations! Your manuscript is now with our production department. 

Kind regards, 

on behalf of

Dr. Nicholas Aderinto Oluwaseyi 

Academic Editor

PLOS ONE